# Iron Deposits in Periaqueductal Gray Matter Are Associated with Poor Response to OnabotulinumtoxinA in Chronic Migraine

**DOI:** 10.3390/toxins12080479

**Published:** 2020-07-28

**Authors:** Clara Domínguez Vivero, Yago Leira, Marta Saavedra Piñeiro, Xiana Rodríguez-Osorio, Pedro Ramos-Cabrer, Carmen Villalba Martín, Tomás Sobrino, Francisco Campos, José Castillo, Rogelio Leira

**Affiliations:** 1Department of Neurology, Headache Unit, Hospital Clinico Universitario, Universidade de Santiago de Compostela, 15706 Santiago de Compostela, Spain; clara.dominguez-vivero@gbhi.org (C.D.V.); marta.saavedra.pineiro@sergas.es (M.S.P.); Xiana.Rodriguez.Osorio@sergas.es (X.R.-O.); 2UCL Eastman Dental Institute and NIHR UCLH Biomedical Research Centre, University College London, London WC1E 6BT, UK; y.leira@ucl.ac.uk; 3Medical-Surgical Dentistry (OMEQUI) Research Group, Health Research Institute of Santiago de Compostela, 15704 Santiago de Compostela, Spain; 4Magnetic Resonance Imaging, Molecular Imaging Unit, CIC biomaGUNE, 20018 Donostia-San Sebastian, Spain; pramos@cicbiomagune.es; 5Ikerbasque, The Basque Foundation for Science, 48013 Bilbao, Spain; 6Department of Radiology, Hospital Clínico Universitario, Universidade de Santiago de Compostela, 15706 Santiago de Compostela, Spain; Carmen.Villalba.martin@sergas.es; 7Clinical Neurosciences Research Laboratory, Health Research Institute of Santiago de Compostela, 15706 Santiago de Compostela, Spain; tomas.sobrino.moreiras@sergas.es (T.S.); francisco.campos.perez@sergas.es (F.C.); rogelio.leira@usc.es (J.C.)

**Keywords:** periaqueductal gray matter, iron deposits, onabotulinumtoxinA, chronic migraine

## Abstract

Previous studies have reported increased brain deposits of iron in patients with chronic migraine (CM). This study aims to determine the relation between iron deposits and outcome after treatment with OnabotulinumtoxinA (OnabotA). Demographic and clinical data were collected for this study through a prospective cohort study including 62 CM patients treated with OnabotA in the Hospital Clínico Universitario de Santiago de Compostela (Spain). Demographic and clinical variables were registered. Selected biomarkers in plasma during interictal periods (calcitonin gene-related peptide (CGRP) and pentraxin-3 (PTX3)) and neuroimaging changes (iron deposits in the red nucleus (RN), substantia nigra (SN), globus pallidus (GP), and periaqueductal gray matter (PAG), and white matter lesions (WML)) were determined. Subjects were classified in responders (≥50% reduction in headache days) or non-responders (<50%). Responders to treatment were younger (mean age difference = 12.2; 95% confidence interval (CI): 5.4–18.9, *p* = 0.001), showed higher serum levels of CGRP (≥50 ng/mL) and PTX3 (≥1000 pg/mL) and smaller iron deposits in the GP and PAG (mean difference = 805.0; 95% CI: 37.9–1572.1 μL, *p* = 0.040 and mean difference = 69.8; 95% CI: 31.0–108.6 μL, *p* = 0.008; respectively). Differences in PAG iron deposits remained significant after adjusting for age (mean difference = 65.7; 95% CI: 22.8–108.6 μL, *p* = 0.003) and were associated with poor response to OnabotA after adjustment for clinical and biochemical variables (odds ratio (OR) = 0.963; 95% CI: 0.927–0.997, *p* = 0.041). We conclude that larger PAG iron deposits are associated with poor response to OnabotA in CM.

## 1. Introduction

The pathophysiology of migraine includes both vascular and neural mechanisms [1]. It involves nociceptive inputs from the raphe and locus coeruleus nuclei [2], the cortical spreading depression (CSD) phenomenon, and trigeminovascular (TGV) system activation [3]. Inflammatory vasoactive peptides promote dilatation of the meningeal vessels, modulate endothelial function [4,5], and could induce blood–brain barrier (BBB) disruption during migraine attacks [6].

The International Headache Society (IHS) recognizes two types of migraine according to their frequency: episodic migraine and chronic migraine (CM). CM is defined as headache occurring on 15 or more days per month for more than 3 months that has the features of migraine headache on at least 8 days per month [7].

OnabotulinumtoxinA (OnabotA) is an effective, safe, and well-tolerated prophylactic treatment for adults with CM [8]. OnabotA acts on C-meningeal fibers inhibiting mechanical nociception [9]. Its mechanism of action is based on the inhibition of soluble N-ethylmaleimide-sensitive fusion attachment protein receptor (SNARE)-mediated vesicle trafficking by interfering with an attachment protein (SNAP-25). This prevents exocytosis of pro-inflammatory and excitatory neurotransmitters and neuropeptides such as substance P, calcitonin gene-related peptide (CGRP), and glutamate from trigeminal terminals. OnabotA also partially blocks the insertion of peripheral pain receptors in the cell membrane and decreases the insertion of pain-sensitive ion channels such as transient receptor potential cation channel subfamily V member 1 (TRPV1) [10].

Several studies have demonstrated the efficacy of OnabotA in the prevention of CM, both in clinical trials [8,11,12,13] and real-life studies subsequently published [14,15,16,17,18,19,20,21,22]. It is known that 70–80% of patients with CM show an improvement with this treatment (defined as a reduction in migraine attack frequency headache days by at least 50% within 3 months), leading to a significantly better functioning of the patients and overall quality of life [23,24]. However, in clinical practice, about 20–30% of patients with CM do not respond to OnabotA. Different studies have tried to find predictors of response to treatment with OnabotA either through clinical features [22,25,26,27,28,29,30,31,32], neuroimaging changes [33,34], or molecular biomarkers [35,36,37]. Some biomarkers related with TGV activation, such as CGRP [35,36] or endothelial dysfunction, such as pentraxin-3 (PTX3) [37,38], have recently been described as predictors of good response to OnabotA in CM (measured in peripheral blood during interictal periods). With regard to imaging features, a recent study has found that responders to OnabotA show cortical thickening in the somatosensory cortex, anterior insula, left superior temporal gyrus, and pars opercularis [33]. As patients with migraine are at higher risk of having white matter lesions (WML), Bumb et al. [34] evaluated them as predictors of outcome after OnabotA with no significant findings. Recently, iron deposits have been reported to be larger in deep brain nuclei in CM patients by our group [39], but their value as predictors of response to treatment with OnabotA has not been examined.

Our aim was to evaluate the association between these neuroimaging changes previously described in CM patients—increased iron deposits in red nucleus (RN), substantia nigra (SN), globus pallidus (GP), and periaqueductal gray matter (PAG) as well as white matter lesions (WML)—and the efficacy of OnabotA in CM.

## 2. Results

### 2.1. Characteristics of the Sample

#### 2.1.1. Demographics

The baseline information on patients with CM (n = 62) treated with OnabotA is shown in Table 1.

Non-responders were significantly older than responders (mean age difference = 12.2; 95% confidence interval (CI): 5.4–18.9, *p* = 0.001).No statistically significant differences were observed regarding gender, body mass index, smoking habit, medication, or time of evolution of chronic migraine.

#### 2.1.2. Characteristics of Migraine

Intensity (mean visual analogic scale (VAS) difference = 0.4; 95% CI: 0.5–1.5, *p* = 0.242), duration (mean difference in hours = 3.7; 95% CI: 18.2–25.6, *p* = 0.554), and frequency (mean difference in days/month = 4.7; 95% CI: 0.08–9.3, *p* = 0.127) of migraine attacks were similar in both responders and non-responders.No differences were found for the presence of aura, allodynia, or tension-type headache.Within the group of responders, 36 (76.6%) were categorized as moderate responders while 11 (23.4%) showed an excellent response to OnabotA.

### 2.2. Predictors of Response

#### 2.2.1. Molecular Biomarkers

Thirty-eight out of 47 (80.9%) chronic migraineurs showing good response to OnabotA presented significantly higher serum levels of CGRP (≥50 ng/mL) compared to 4 out of 15 (26.7%) with a poor outcome.Similarly, 87.2% (41/47) of responders had elevated serum levels of PTX3 (≥1000 pg/mL) in comparison to 20.0% (3/15) of non-responders.

#### 2.2.2. Imaging Biomarkers

Statistically significant differences between responders and non-responders were found for iron deposition in the GP and PAG (mean difference = 805.0; 95% CI: 37.9–1572.1 μL, *p* = 0.040 and mean difference = 69.8; 95% CI: 31.0–108.6 μL, *p* = 0.008; respectively). Adjustment for age in the multivariate model changed statistical significance for GP (mean difference = 472.4; 95% CI: 341.5–1286.4 μL, *p* = 0.250) but not for PAG (mean difference = 65.7; 95% CI: 22.8–108.6 μL, *p* = 0.003).No discrepancies were observed for the prevalence, number, and location of WML between responders and non-responders (Table 2).Iron deposition in the PAG was associated with higher odds of poor response to OnabotA (Figure 1). A 10% increase in iron ground volumes in the PAG was associated with an odds ratio for poor response to treatment of 0.973 (95% CI: 0.955–0.991, *p* = 0.040) independently of age and GP (Figure 1, Model I; Table 3).After adjustment for biochemical variables, larger iron deposits in the PAG remained significantly associated with poor response to OnabotA (odds ratio (OR) = 0.963; 95% CI: 0.927–0.997, *p* = 0.041) (Figure 1, Model II; Table 3).

## 3. Discussion

In this study, we found an association between larger T2 hypointense voxels consistent with increased iron levels in the PAG and poor response to OnabotA treatment in CM patients. This association remained significant after adjustment for clinical variables and serum biomarker levels. Our findings suggest that iron deposits in the PAG could work as neuroimaging predictors of outcome after treatment with OnabotA in CM.

OnabotA is an effective preventive treatment for patients with CM; however, about one third of patients do not respond to it, even after several attempts [22]. While the first PREEMPT trials could not find any predictor of response [11,12], later reports have shown several clinical features that can predict efficacy such as age [22], time of evolution of migraine [22,26,31], unilateral pain [25], imploding headache [27], allodynia [25], comorbid depression, and medication overuse [28,30]. Studies addressing potential predictors of response to the only other treatment approved to prevent CM (topiramate) are scarce. Interestingly, some of the predictors reported are similar to those of OnabotA: negative previous experience with other prophylactic therapy, chronic daily headache (CDH), and, most notably, CDH of more than 6 months duration [40,41]. Time of evolution of CM seems able to predict response to both preventive therapies.

Regarding molecular biomarkers, interictal peripheral blood levels of GCRP and PTX3 have been reported as predictors of response to OnabotA in CM [35,36,37]. It is clear that CGRP is clearly involved in migraine pathophysiology and in related mechanisms such as pain transmission, inflammation, and vasodilation. PTX3 is related with local endothelial dysfunction, but its exact role in migraine is still unknown. To date, few studies have focused on imaging predictors of outcome after treatment with OnabotA. Hubbard et al. [33] found increased cortical thickness in the right primary sensory cortex, anterior insula, left superior temporal gyrus, and pars opercularis in responders to treatment whereas Bumb et al. [34] could not find an association between white matter burden and efficacy of OnabotA. In our study, findings related to WML were similar: no association between load of WML and response to treatment. There are no studies regarding molecular or imaging predictors of response to other approved preventive treatments (topiramate).

In this study, an association was found between larger iron deposits in the PAG and a poor response to OnabotA in CM patients. These findings could be the neuroimaging correlate of clinical predictors of response such as duration of disease, frequency of attacks, or allodynia. The PAG is an essential modulator of pain that contributes to central sensitization and development of secondary hyperalgesia [42]. The PAG’s structure includes various layered neurons around the aquaeductus mesencephali [39] and works as a hub in the pain-processing network. It conveys powerful descending antinociceptive functions and ascending connections with diencephalic and cortical structures involved in pain processing [43]. Some studies have shown how PAG activation is modulated by expectation of pain [44] and placebo analgesia [45].

Due to its important role in pain regulation, PAG dysfunction has been proposed as one of the underlying mechanisms of migraine [46], and particularly, of migraine chronification. Several studies have reported iron deposition in the PAG, together with some diencephalic structures, in patients with CM and daily headache [39,47,48,49]. Some other relevant reports, such as the CAMERA study [48], did not provide data supporting the hypothesis of increased iron accumulation in deep brain nuclei in patients with migraine, although the iron ground volumes in the PAG were not specifically evaluated in any of the CAMERA studies [48,49]. A previous study by this research group [39] found larger iron deposits in the PAG in CM when compared to episodic migraine (EM) and in EM when compared to healthy controls, suggesting a progressive increase in structural damage as migraine frequency and severity increases. Iron deposits in the PAG and basal ganglia were also correlated with time from diagnosis of migraine.

The mechanism by which iron deposits increase in the PAG of CM patients is still unknown and is probably not related with OnabotA’s mechanism of action. In the human brain, iron is stored as ferritin mainly in the myelin sheets. Iron levels increase physiologically with age and oxidative stress, and it can amplify oxidative damage [50]. As migraine attacks activate PAG [43], we could hypothesize that repeated episodes of migraine increase free-radical release in the area and contribute to iron deposition. Inflammation has a role in cellular death and destruction mediated by iron accumulation [50] and, therefore, chronic insult to the PAG by repeated neural and vascular inflammatory mechanisms during pain may lead to the enlargement of iron deposits as well. Finally, hyperoxia (secondary to vasodilation) can release free radicals and harm cells causing iron sequestration in this tissue [51]. Endothelial dysfunction and BBB disruption favor leakage between cerebrospinal fluid (CSF) and blood. This leakage may also contribute to inflammation and increase iron accumulation. Brain endothelial cells are iron repositories, and their activation could modify iron balance in affected brain areas [52]. These mechanisms have already been reported in other neuroinflammatory brain diseases [53,54,55]. In summary, iron deposits may be a consequence of a higher activation rate of pain-regulation structures due to repeated episodes of pain and, therefore, work as a neuroimaging correlate of migraine duration and frequency of attacks.

It is also arguable that higher iron deposition could be the first event and the cause of an increased susceptibility to pain and poor response to treatment. According to this hypothesis, dysfunction of brain endothelial cells, which act as reservoirs of iron, could lead to iron accumulation in certain structures such as the PAG [52,56,57,58,59]. Although OnabotA exerts its action in the periphery, it is possible that once central structures such as the PAG are damaged, its effects are no longer noticeable.

Previous literature suggests that the severity of migraine, measured by clinical features such as longer time of evolution [22], allodynia [25], bilateral pain [25], or age, is related with worse outcomes after treatment with OnabotA. In addition, several studies have reported structural changes in the PAG and other nuclei related with pain processing. Our results add to previous findings, showing that structural imaging changes that are related with disease burden can also predict response to treatment with OnabotA, independently of clinical variables and molecular biomarkers. These findings reinforce the idea that a longer and more severe disease entails structural changes in the central networks of pain. Specifically, once damage is established in the brainstem or higher structures involved in pain regulation, OnabotA’s effects on peripheral terminals may be no longer effective. Larger studies including more patients and longer follow-up periods are needed in order to determine the clinical utility of these imaging predictors. Clinical factors, molecular markers, and imaging features could even be combined in a tool to determine the individual chances of response to treatment.

Our study has several limitations. First, age has been shown to have a great influence in iron deposition in subcortical nuclei [60,61,62], although results regarding iron deposition in the PAG and age are more inconsistent [63]. Nevertheless, our analyses have been adjusted by age. Our sample is mainly composed of females, and iron metabolism has been proved to be different in both sexes [64]. However, as most patients with CM are female, our sample reflects the characteristics of the general population with migraine. An important limitation of our study is the short follow-up period after OnabotA treatment, as previous reports have shown that efficacy can change greatly after 2–3 cycles of treatment [18]. Nonetheless, the majority of patients show some kind of response after the first dose of OnabotulinumtoxinA, and a 12-week period of follow-up has been used in previous studies evaluating response to OnabotulinumtoxinA. Further studies evaluating long-term response to treatment are needed. Some clinical variables that have been reported to predict efficacy of OnabotA were not used, such as laterality of pain or imploding characteristics of pain. We used headache diaries to register headache days after treatment, but baseline frequency is based on patient recall, which could bias our results. Finally, preventive treatments used concurrently with OnabotA were not considered in the analysis, and they could have influenced our results.

## 4. Conclusions

Our study shows for the first time that iron deposits in the PAG in CM patients can predict response to treatment with OnabotA. Our findings are independent of molecular predictors previously related with treatment outcome with OnabotA. Further research with larger samples and follow-up is needed to determine causality. Our results contribute to the increasing body of evidence regarding biomarkers of chronification and predictors of response to OnabotA.

## 5. Materials and Methods

### 5.1. Study Protocol

This is an observational study. This study uses a subgroup of patients and their magnetic resonance imaging (MRI) data from a previous study [39]. Subjects were recruited prospectively among visitors of the outpatient Headache Clinic of Department of Neurology, at Hospital Clínico Universitario de Santiago de Compostela between January 2014 and June 2015. Sixty-two subjects diagnosed with CM according to International Classification of Headache Disorders, 3rd edition criteria [7] were selected. These subjects were all candidates to receive treatment with OnabotA in our country, where treatment with OnabotA can be started once a patient with CM has already been using in a proper way two preventive medications from two different pharmaceutical groups with poor efficacy and/or tolerability. Eligible subjects were invited to perform a magnetic resonance imaging (MRI) examination and blood analyses. All subjects were older than 18 years. Clinical variables were recorded, including demographic data (age, gender) and personal and family history, imaging studies were performed at 3T MRI, and selected molecular makers were determined in peripheral blood.

Subjects were excluded if they had any of the following criteria: (1) high blood pressure (known high blood pressure or >2 measurements greater than 140/90 mm Hg); (2) coronary disease; (3) diabetes mellitus; (4) hypercholesterolemia (pharmacologically treated or fasting serum cholesterol >200 mg/dL); (5) infectious diseases; (6) chronic inflammatory conditions such as rheumatoid arthritis, inflammatory bowel disease, systemic lupus, and other autoimmune conditions; (7) severe systemic diseases; (8) oligomenorrhea, polymenorrhea, or polycystic ovarian syndrome; (9) pregnancy or lactation; (10) obesity (body mass index >30 kg/m^2^); (11) smoking habit (within the previous 12 months); (12) recent consumption of vasoactive drugs (>4 times the medium half-life of the active substance).

OnabotA was administered following the PREEMPT protocol [11] without a “follow the pain” strategy-155 International Units (IU) in 31 pericranial injection sites- every 12 weeks. Treatment with other prophylactic drugs was allowed.

Patients used diaries to record the number of episodes of moderate–severe acute headache lasting longer than 4 h during the 12 weeks following treatment with OnabotA. Time of evolution of CM was defined as months since CM diagnostic criteria were fulfilled. We classified subjects in two groups according to their response: non-responders (<50% reduction in frequency of headache) and responders (≥50% reduction in frequency of headache). We divided responders into two subgroups: moderate responders (50–75% reduction in frequency of headache) and excellent responders (>75% reduction in frequency of headache) [36].

### 5.2. Laboratory Tests

Blood samples were drawn before 11 a.m. after a 12 h fasting period in our clinic. They were collected in chemistry test tubes, centrifuged at 3000× *g* for 15 min, and immediately frozen and stored at −80 °C. Serum levels of PTX3 (Assay Biotech, Sunnyvale, CA, USA), and CGRP (Phoenix Pharmaceuticals, Burlingame, CA, USA) were measured using commercial ELISA kits following the manufacturer’s instructions. The intra-assay and inter-assay coefficients of variation (CV) for all molecular markers were <8%. Determinations were performed in a laboratory blinded to clinical data.

### 5.3. Neuroimaging Variables

Images were acquired in a 3 Tesla Philips Achieva system. The protocols of MRI acquisition and image analysis have been previously published [39]. MRI studies were performed by a radiologist (C.V.) and a physicist (P.R.) blinded to clinical data using self-developed routines for the NIH (National Institute of Health) software platform ImageJ [65] according to a modified version of Jurgens’s methodology [66]. A region of interest (ROI) in the corpus callosum was identified in each subject, and we determined its average signal intensity. Using the total set of images (3836 MR images) we built a histogram of pixel intensity and adjusted the corresponding plot to 4 independent Gaussian functions. After this, pixels of other brain areas were classified according to their intensity values and the defined thresholds. Hypo-intense areas were those pixels with intensities of 0–65, and white matter areas had intensities ranging from 65 to 140. Pixels with intensities of 140–210 were identified as grey matter, and those with intensities ranging from 210 to 255 as CSF. RN, GP, and PAG (defined manually as a circular region of 4 mm diameter around the aqueduct of Sylvius) were explored to look for hypo-intense areas counting the total number of hypo-intense pixels, as well as mean value and standard deviation of the signal intensities. T2 sequences were used to identify the number and location of WMLs by a radiologist (C.V.).

### 5.4. Standard Protocol Approvals, Registrations, and Patient Consents

The Research Ethics Committee of University Clinical Hospital of Hospital Santiago de Compostela (Spain) approved the study (ID: 2016/085; approval date: 29 March 2016). The study conforms with the World Medical Association Declaration of Helsinki. All subjects in this study provided signed informed consent.

### 5.5. Statistical Analysis

Continuous normally distributed variables (Kolmogorov–Smirnov test) were reported as mean ± standard deviation (SD), whereas continuous non-normally distributed variables were expressed as median [P_25_, P_75_]. Mean difference (95% CI) was also used to express continuous variables. Categorical variables were reported as percentages (%). Differences between two groups were assessed by independent *t* test (continuous normally distributed variables), Mann–Whitney test (continuous non-normally distributed variables), and *X*^2^ test (categorical variables). The p-values for comparisons between different categories of categorical variables were calculated as follows: (i) adjusted z-score were obtained for each value, (ii) *X*^2^ was calculated for each value by using the formula (adjusted z-score × adjusted z-score), (iii) p-values were obtained by applying the numeric expression: SIG.CHISQ (*X*^2^, 1 degree of freedom), and finally (iv) p-values were compared to Bonferroni correction post hoc test to confirm significance. Analysis of covariance (_ANCOVA_) was done creating adjusted models using age as covariate to compare mean values of iron deposition in different locations between responders and non-responders. Logistic regression models were created to test potential associations between iron volume and OnabotA efficacy (responders versus non-responders), adjusted for statistically significant variables in the univariate analysis or previously published predictors of efficacy to this treatment approach [39]. A correlation matrix confirmed absence of multicollinearity between predictors. All two-tailed tests were performed at a significance level of α = 0.05. All data analyses were performed with IBM SPSS Statistics 20.0 software for Mac (SPSS Inc., Chicago, IL, USA).

## Figures and Tables

**Figure 1 toxins-12-00479-f001:**
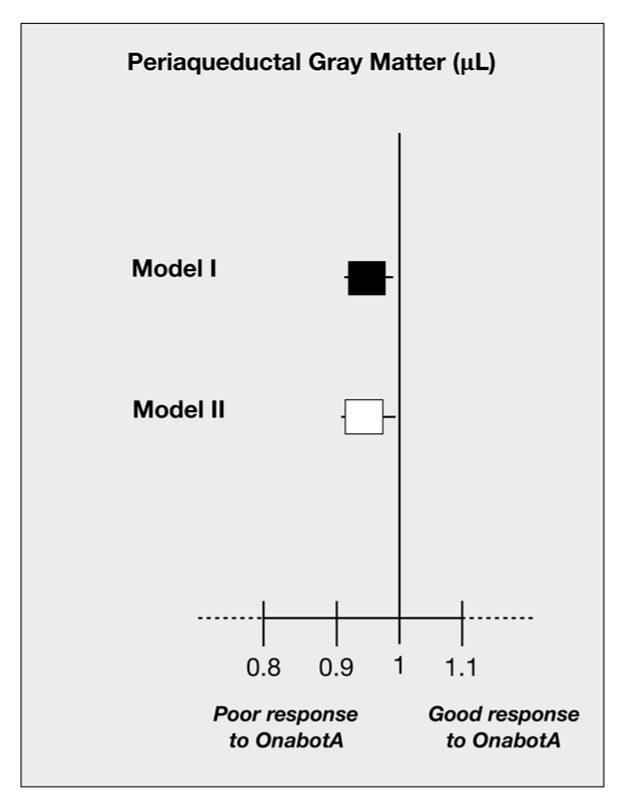
Odds ratio (95% confidence intervals (CIs)) of iron deposition in periaqueductal gray matter and poor response to OnabotulinumtoxinA (OnabotA). Black squares represent odds ratio adjusted for age and globus pallidus (Model I). White squares represent odds ratio adjusted for age, calcitonin gene-related peptide (CGRP) serum levels ≥50 ng/mL and pentraxin-3 (PTX3) serum levels ≥1000 pg/mL (Model II).

**Table 1 toxins-12-00479-t001:** Baseline characteristics of patients treated with OnabotA (n = 62).

Variables	Responders (n = 47)	Non Responders (n = 15)	*p*-Value
**Age (years)** **Gender, Females, n (%)**	39.4 ± 12.0 46 (97.9)	51.6 ± 9.1 14 (93.3)	**0.001** 0.428
**Body mass index (kg/m^2^)**	24.8 [22.6, 27.4]	25.5 [23.7, 28.9]	0.425
**Allodynia, n (%)**	15 (31.9)	8 (53.3)	0.135
**Aura, n (%)**	22 (46.8)	9 (60.0)	0.554
**Tension-type headache, n (%)**	25 (53.2)	10 (66.7)	0.537
**Preventive treatment (≥2 drugs), n (%)**	24 (51.0)	8 (53.4)	0.117
**Symptomatic treatment (≥2 drugs), n (%)**	29 (61.7)	8 (53.3)	0.749

Significant results are reported in bold.

**Table 2 toxins-12-00479-t002:** Neuroimaging outcomes according to treatment response.

Neuroimaging Variables	Responders (n = 47)	Non Responders (n = 15)	*p*-Value
**Iron deposits (μL)**			
Red Nucleus (median [interquartile range])	39.6 [3.5, 99.0]	83.7 [19.3, 128.3]	0.244
Substantia Nigra (median [interquartile range])	205.6 [105.0, 397.4]	257.4 [158.8, 607.0]	0.305
Globus Pallidus (mean ± standard deviation)	1690.4 ± 995.5	2495.5 ± 1852.3	**0.040**
Periaqueductal Gray Matter (median [interquartile range])	352.0 [265.2, 365.7]	455.5 [408.5, 473.5]	**0.008**
**Presence of White Matter Lesions, n (%)**	24 (51.1)	11 (73.3)	0.130
**Number of White Matter Lesions**			
<3, n (%)	3 (12.5)	0. (0.0)	0.230
3–6, n (%)	13 (54.2)	7 (63.6)	0.620
>6, n (%)	8 (33.3)	4 (36.4)	0.840
**Location of White Matter Lesions**			
Subcortical, n (%)	8 (33.3)	4 (36.4)	0.860
Subcortical + periventricular, n (%)	11 (45.8)	7 (63.6)	0.330
Subcortical + other locations, n (%)	5 (20.8)	0 (0.0)	0.100

Significant results are reported in bold.

**Table 3 toxins-12-00479-t003:** Logistic regression analysis.

	OR	95% CI	*p*-Value
**Model I**			
Age	0.882	0.769–0.970	**0.012**
Globus Pallidus (µL)	0.999	0.997–1.002	0.613
Periaqueductal gray (µL)	0.973	0.955–0.991	**0.040**
**Model II**			
Age	0.815	0.668–0.995	**0.044**
CGRP ≥ 50 (ng/mL)	1.026	1.001–1.050	**0.034**
PTX3 ≥ 1000 (ng/mL)	1.008	1.001–1.016	**0.037**
Periaqueductal gray (µL)	0.963	0.927–0.997	**0.041**
Dependent variable: OnabotA response (good versus poor)			

CI: confidence interval; OR: odds ratio; OnabotA: onabotulinumtoxinA; CGRP: calcitonin gene-related peptide; PTX3: pentraxin-3. Significant results are reported in bold.

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
