# Peer review of "Iron Deposits in Periaqueductal Gray Matter Are Associated with Poor Response to OnabotulinumtoxinA in Chronic Migraine"

_toxins, 2020, doi:10.3390/toxins12080479_

Round 1
Reviewer 1 Report
This manuscript describes a novel finding of a iron deposits in PAG as a predictor of response of chronic migraine patients to Onabotulinum toxin A treatment. Overall, the study is very well described, well written, and captivating. The results section seems somewhat short relative to the discussion and the amount of work described. I would suggest to include the supplemental table into the results and write the text in small paragraphs rather than bullet points, however, that is a style choice suggestion and not a concern that must be addressed.
It is intriguing to me that the BBB can get damaged during CM in some cases, as BoNTs don't cross the BBB and we do not understand the mechanism by which BoNTs actually improve CM although this doesn't explain the findings presented in this paper. The manuscript would benefit from at least a short paragraph describing our current state of knowledge of how BoNT may act to reduce migraine headaches.
Author Response
POINT 1. This manuscript describes a novel finding of a iron deposits in PAG as a predictor of response of chronic migraine patients to Onabotulinumtoxin A treatment. Overall, the study is very well described, well written, and captivating. The results section seems somewhat short relative to the discussion and the amount of work described. I would suggest to include the supplemental table into the results and write the text in small paragraphs rather than bullet points, however, that is a style choice suggestion and not a concern that must be addressed.
Thank you for your suggestions. We have moved the supplementary table to Results section, where it is now complementing the information provided by Figure 1. Regarding the formatting of our results, the bullet point choice is due to a style suggestion by the Journal, and therefore we have decided to keep it that way.
POINT 2: It is intriguing to me that the BBB can get damaged during CM in some cases, as BoNTs don't cross the BBB and we do not understand the mechanism by which BoNTs actually improve CM although this doesn't explain the findings presented in this paper. The manuscript would benefit from at least a short paragraph describing our current state of knowledge of how BoNT may act to reduce migraine headaches.
Thank you for your comment. We have introduced a short explanation about OnabotA mechanism of action in the Introduction and also some sentences in the Discussion remarking that there is probably not a relation between OnabotA mechanism of action and iron deposition. It is, indeed, still intriguing, the particular mechanisms of action of OnabotA and the differences between patients in some physiopathological features, such as BBB disruption. In this work, we found that iron deposits can predict response to OnabotA, probably not because there is any relation between OnabotA mechanism of action and iron deposition, but because iron deposition is related with repeated episodes of pain and migraine chronification.
LINES 40-48: “OnabotulinumtoxinA mechanisms of action is based on the inhibition of soluble N-ethylmaleimide-sensitive fusion attachment protein receptor (SNARE)-mediated vesicle trafficking by interfering with an attachment protein (SNAP-25). This prevents exocytosis of pro-inflammatory and excitatory neurotransmitters and neuropeptides such as substance P, calcitonin gene-related peptide, and glutamate from trigeminal terminals, and also blocks, at least partially, the membrane insertion of peripheral pain receptors. OnabotulinumtoxinA also decreases the insertion of pain-sensitive ion channels such as transient receptor potential cation channel subfamily V member 1 (TRPV1)”.
LINES 182-183: “The mechanism by which iron deposits increase in PAG of CM patients is still unknown, and is probably not related with OnabotA mechanism of action (…)”
LINES 274-275: “Although OnabotA exerts its action in the periphery, it is possible that once central structures such as PAG are damaged, its effects are not longer noticeable.”
Reviewer 2 Report
Abstract
Line 6 - please explain the abbreviation CM
Materials and methods
Line - 216-219 The sentence is not to be clearly understood.
Line - 241 when were the blood samples collected?
It would be interesting to state the exact age of the patients.
How exactly were the patients treated (duration, dose)?
Results
Line 70 - You wrote: no statistically significant differences were observed regarding: gender ...
but were both men and women examined in this study? There is no clear indication of this (not even in materials and methods).
Author Response
POINT 1. Abstract. Line 6 - please explain the abbreviation CM
Thanks for your comment. We have added “chronic migraine” to the text.
POINT 2. Materials and methods. Line - 216-219 The sentence is not to be clearly understood.
It is true that the sentence can be confusing. We have rephrased so that it is clear that subjects were diagnosed with CM and were also candidates to receive treatment with OnabotA according to Spanish guidelines.
LINE 315. “Sixty-two subjects diagnosed with CM according to International Classification of Headache Disorders, 3rd edition criteria [7] were selected. They were all candidates to receive treatment with OnabotA in our country (…)”
POINT 3. Line - 241 when were the blood samples collected?
We have added this information under section 5.2, LINE 385: “Blood samples were drawn before 11 am after a 12 hour fasting period in our clinic.”
POINT 4. It would be interesting to state the exact age of the patients.
Thank you for acknowledging this. We have added the age of patients in Table 1.
POINT 5. How exactly were the patients treated (duration, dose)?
We have added this information in the Methods Section, LINE 375 as “(155 UI in 31 pericranial injection sites) every 12 weeks”
POINT 6. Results. Line 70 - You wrote: no statistically significant differences were observed regarding: gender ...but were both men and women examined in this study? There is no clear indication of this (not even in materials and methods).
Thank you for your question. As you can see in Table 1, there was 1 man among responders and 1 man among non-responders. This corresponds with the ample predominance of women in CM in clinical practice. There was no significant difference between groups regarding the percentage of women (97.9% vs. 93.3%)
Reviewer 3 Report
I have a few questions/doubts, mainly regarding overlap with the previous publication from the group (Neurology 2019 Mar 5;92(10):e1076-e1085):
- Where the same patients with CM investigated in the current study? Do the same MRI exams were used for the current study?
- The discussion is in part quite similar to the previous paper, mainly on MRI findings in CM
- However, the new aspect of prediction of response to Botulinum Neurotoxin is only barely discussed in the current manuscipt (ln. 177-188)
One very large limitation of the current study, reported from the authors themselves, is the fact that therapy response was evaluated after only one injection cycle. Based on this evaluation the patients were separated in the two groups.
Furthermore, the group of non-responders is small.
The authors should report/discuss predictors of treatment response to other prophylactic (oral drug) treatment.
Why is such a latency (of 5 years) between recruitment of patients and submission of the manuscript?
Some small typos:
Ln. "Fibers" instead of "fibbers"
Ln. 139 "hyperalgesia" instead of "hiperalgesia"
Author Response
POINT 1. I have a few questions/doubts, mainly regarding overlap with the previous publication from the group (Neurology 2019 Mar 5;92(10):e1076-e1085):Where the same patients with CM investigated in the current study? Do the same MRI exams were used for the current study?
Thank you for the question. Yes, this study uses a subgroup of patients from the previous published study, those that were candidates to receive treatment with OnabotA. The MRI exams were the same performed for the previous study.
POINT 2.The discussion is in part quite similar to the previous paper, mainly on MRI findings in CM. However, the new aspect of prediction of response to Botulinum Neurotoxin is only barely discussed in the current manuscipt (ln. 177-188)
Thanks for your comment. There are some similarities in the discussion as this summarizes what we actually know about iron deposits in CM. However, we have modified it and rephrased it, trying to emphasize the value of our findings regarding prediction of response to treatment. We have also introduced more information on OnabotA mechanism of action.
POINT 3. One very large limitation of the current study, reported from the authors themselves, is the fact that therapy response was evaluated after only one injection cycle. Based on this evaluation the patients were separated in the two groups.Furthermore, the group of non-responders is small.
Thank you for your comment. We agree that this is an important limitation of the study. Data regarding rate of response after a longer period should have been gathered, however, the standard evaluation of efficacy in OnabotA is done after 12 weeks of treatment, although studies based on RWD have shown that some non-responders can respond later. Regarding the rate of non-responders, it corresponds with the rates of response found in other studies.
POINT 4. The authors should report/discuss predictors of treatment response to other prophylactic (oral drug) treatment.
Thank you for your suggestion. We have added a brief discussion on predictors of the only other preventive treatment approved for CM, topiramate.
LINES 145-150: “Studies addressing potential predictors of response to the other CM preventive approved treatment, topiramate, are scarce. Interestingly, some of the predictors reported are similar to those of OnabotA: negative previous experience with other prophylactic therapy, chronic daily headache (CDH), and, most notably, CDH of more than 6 months duration (40,41). Time of evolution of CM seems able to predict response to both preventive therapies.”
POINT 5. Why is such a latency (of 5 years) between recruitment of patients and submission of the manuscript?
Thank you for your question. This is a subanalysis of previous data focused on molecular and neuroimaging biomarkers in CM. Once these results were analysed and published we decided to evaluate the possible value of these biomarkers as predictors of response. This is the main reason for the delay in publishing these results.
POINT 6. Some small typos: Ln. "Fibers" instead of "fibbers" Ln. 139 "hyperalgesia" instead of "hiperalgesia"
Thanks, typos have been corrected.
Round 2
Reviewer 3 Report
I thank the authors for their corrections and modifications.
I would suggest, that it has to be stated, that this study uses a subgroup of patients and their MRI data from their previous study.
Author Response
Dear reviewer.
Thank you for your suggestion. We have tried to address your comments in the best way possible.
We have added a sentence stating that the study uses a subgroup of patients from the previous one (Reference 39). We included this sentence o=in Materials and Methods, section 5.1 (Study Protocol), lines 222-223
"This study uses a subgroup of patients and their MRI data from a previous study (39)".
We have also had the paper reviewed by a native speaker and have implemented some minor changes.
We hope you find these changes satisfactory.
Again thanks for your time and effort.